# Pilot study of Tremelimumab with and without cryoablation in patients with metastatic renal cell carcinoma

Matthew T. Campbell [1], Surena F. Matin [2], Alda L. Tam[3], Rahul A. Sheth [3], Kamran Ahrar[3], Rebecca S. Tidwell [4], Priya Rao[5], Jose A. Karam [2,6], Christopher G. Wood[2], Nizar M. Tannir[1], Eric Jonasch [1], Jianjun Gao [1], Amado J. Zurita[1], Amishi Y. Shah [1], Sonali Jindal[7], Fei Duan[7], Sreyashi Basu [7], Hong Chen[7], Alexsandra B. Espejo [7], James P. Allison[7,8], Shalini S. Yadav[7] & Padmanee Sharma [1,7,8✉]

Cryoablation in combination with immune checkpoint therapy was previously reported to improve anti-tumor immune responses in pre-clinical studies. Here we report a pilot study of anti-CTLA-4 (tremelimumab) with (n = 15) or without (n = 14) cryoablation in patients with metastatic renal cell carcinoma (NCT02626130), 18 patients with clear cell and 11 patients with non-clear cell histologies. The primary endpoint is safety, secondary endpoints include objective response rate, progression-free survival, and immune monitoring studies. Safety data indicate ≥ grade 3 treatment-related adverse events in 16 of 29 patients (55%) including 6 diarrhea/colitis, 3 hepatitis, 1 pneumonitis, and 1 glomerulonephritis. Toxicity leading to treatment discontinuation occurs in 5 patients in each arm. 3 patients with clear cell histology experience durable responses. One patient in the tremelimumab arm experiences an objective response, the median progression-free survival for all patients is 3.3 months (95% CI: 2.0, 5.3 months). Exploratory immune monitoring analysis of baseline and post-treatment tumor tissue samples shows that treatment increases immune cell infiltration and tertiary lymphoid structures in clear cell but not in non-clear cell. In clear cell, cryoablation plus tremelimumab leads to a significant increase in immune cell infiltration. These data high-light that treatment with tremelimumab plus cryotherapy is feasible and modulates the immune microenvironment in patients with metastatic clear cell histology.

[1] Department of Genitourinary Medical Oncology, The University of Texas MD Anderson, Houston, TX, USA. [2] Department of Urology, The University of Texas MD Anderson Cancer Center, Houston, TX, USA. [3] Department of Interventional Radiology, The University of Texas of MD Anderson Cancer Center, Houston, TX, USA. [4] Department of Biostatistics, The University of Texas MD Anderson Cancer Center, Houston, TX, USA. [5] Department of Pathology, The University of Texas MD Anderson Cancer Center, Houston, TX, USA. [6] Department of Translational Molecular Pathology, The University of Texas MD Anderson Cancer Center, Houston, TX, USA. [7] The Immunotherapy Platform, The University of Texas MD Anderson Cancer Center, Houston, TX, USA. [8] Department of Immunology, The University of Texas MD Anderson Cancer Center, Houston, TX, USA. ✉email: padsharma@mdanderson.org

Cytotoxic T lymphocyte antigen 4 (CTLA-4) directly competes against CD28 to abrogate T-cell responses, which led to the development of an antibody targeting CTLA-4 as a therapy to enhance antitumor T-cell activity[1,2]. A phase III study in metastatic melanoma found improved overall survival (OS) for patients who received ipilimumab, an IgG1 monoclonal antibody targeting CTLA-4, which led to FDA approval[3]. Although the median survival was improved for the patients who received ipilimumab, the duration of benefit was profound with ~20% of the patients still alive 10 years after completion of therapy[4]. Tremelimumab, an IgG2 monoclonal antibody targeting CTLA-4, was also tested in a phase III trial in melanoma. Although the response rate and duration of response were numerically similar to the study with ipilimumab, the primary endpoint of improvement of median OS was not achieved[5]. Identifying the subgroup of patients with metastatic melanoma who are likely to be cured with ipilimumab has remained elusive with baseline clinical characteristics. However, when comparing baseline with post ipilimumab treated tumors, an increased presence of CD4 T cells that express high levels of inducible T-cell co-stimulator (ICOS) has been correlated repeatedly with improved patient outcomes across multiple tumor types[6–9].

In contrast to metastatic melanoma, the role of single-agent anti-CTLA-4 therapy is not well defined in metastatic renal cell carcinoma (mRCC). The most common histology in mRCC is clear cell histology (mccRCC) representing 70–75% of patients[10]. The remaining patients with non-clear cell or variant histology (mnccRCC) include papillary (10–15%), chromophobe (5%), unclassified (5%), and more rare subtypes. To date, a single-phase II study of single-agent ipilimumab in mccRCC found an objective response rate of 9.8% (6 out of 61), with noted high-grade immune-related adverse events in a subset of patients[11]. No long-term follow-up is available for the patients treated in this study. At present, little is known about the long-term efficacy of single-agent anti-CLTA-4 therapy in patients with mRCC and there is no available data exploring its activity in patients with mnccRCC.

Efforts to enhance the efficacy of anti-CTLA-4 based therapy, while limiting additional severe immune-related toxicity, have been pursued with the addition of other immune checkpoint inhibitors, radiation, vaccine therapy, targeted therapy, and other approaches. Cryoablation (cryo) refers to a local thermal treatment that employs extreme cold to achieve tumor cell death. For non-metastatic renal cell carcinoma (RCC), cryo has been used for poor surgical candidates with renal masses <4 cm, resulting in excellent local control[12]. In addition, cryo has also been used to palliate symptoms in patients with painful RCC metastases[13]. Cryo has been found to cause both necrotic and apoptotic cell death through a variety of mechanisms while leaving the vasculature largely intact[14]. As a result, cryo has been investigated as a modality that may induce antitumor immune responses in animal models. In an orthotopic RCC rat model, cryoablation led to significant infiltration of the tumor with neutrophils, macrophages, and T cells[15]. In a murine TRAMP-C2 prostate cancer cell line model, the combination of anti-CTLA-4 with cryo led to improved antitumor immune responses and tumor rejection[16]. With this background, our group initiated a pilot study of the anti-CTLA-4 agent, tremelimumab, with or without cryo in patients with mRCC allowing for any histology and any number of prior treatment lines. We hypothesized that the addition of cryoablation to tremelimumab would be safe and would lead to a detectable increase in immune cell infiltration within the tumor microenvironment. Owing to a lack of data with anti-CTLA-4 single-agent therapy in mRCC, the study was histology agnostic allowing accrual of all subtypes.

Here, we show that tremelimumab with or without cryoablation has an acceptable safety profile and leads to an increase in immune cell infiltration and tertiary lymphoid structures (TLS) in mccRCC but not in mnccRCC. In mccRCC, tremelimumab plus cryoablation leads to a significant increase in immune cell infiltration. This study shows that cryoablation in combination with immune checkpoint therapy (ICT) is feasible in RCC.

## Results

**Baseline demographics.** A total of 30 patients were enrolled in the study, underwent randomization, and one patient withdrew consent prior to treatment. Of the 29 patients treated on protocol, 14 were randomized to arm A, tremelimumab-only, and 15 were randomized to arm B, cryoablation plus tremelimumab (cryo-tremelimumab). The trial design is shown in Fig. 1a, consort diagram is shown in Fig. 1b. Table 1 includes baseline demographics including prior treatments and the site used for cryo in Arm B. Of the 29 patients, 24 (83%) were male, 26 (90%) were Caucasian, 17 (59%) had undergone prior nephrectomy, 18 (62%) had clear cell histology. The majority of patients, 20 (68%), had intermediate-risk disease based on the International Metastatic Renal Cell Carcinoma Database Consortium criteria while 4 (18%) and 5 (17%) had good and poor risk, respectively. Twenty patients (68%) did not have prior systemic treatment, 9 (32%) had previous systemic treatment with eight of the nine patients previously treated with a tyrosine kinase inhibitor, and six of the nine patients previously treated with nivolumab. Of note, five of the six patients who received prior treatment with nivolumab were in the cryoablation plus tremelimumab arm.

**Safety.** The trial did not stop early for extreme toxicities since the stopping boundaries were never crossed. The empirical rate for each arm were 3/14 (21%) and 4/15 (27%) for tremelimumab monotherapy and cryoablation plus tremelimumab combination therapy, respectively. The posterior estimates and 95% credible intervals were 22% (6%, 44%) and 26% (9%, 49%), respectively. Grade 3 or higher adverse events at least possibly related to treatment occurred in 16 of 29 (55.2%) patients on study. Immune-related toxicities of interest included grade 3 or higher diarrhea/colitis and occurred in 6 patients (21%), hepatitis in 3 (10%), pneumonitis in 1 (3%), and glomerulonephritis in 1 (3%). Toxicity leading to treatment discontinuation occurred in 5 of 15 (33%) patients in the tremelimumab-only arm and in 5 of 14 (35.7%) patients in the cryo-tremelimumab arm. Toxicities leading to treatment discontinuation included diarrhea/colitis in three patients in both arms, immune-related hepatitis in two patients in the tremelimumab-only arm and one in the combination arm, one patient with pneumonitis, and one patient with immune-related glomerulonephritis in the cryo-tremelimumab arm. Three patients in the tremelimumab arm and four patients in the cryo-tremelimumab arm experienced grade 3 or higher toxicity that did not improve within 2 weeks of high dose corticosteroids and required additional immunosuppressive treatment. One additional patient in the tremelimumab-only arm had recurrent grade 1 colitis and required additional immunosuppressive therapy after failure to wean steroids. No new toxicity signals were generated and cryoablation did not statistically enhance toxicity to anti-CTLA-4 therapy. Treatment-related toxicities grade 3 or higher are shown in Table 2, all toxicities are listed in Supplementary Table 1, and all related toxicities are listed in Supplementary Table 2.

**Surgery.** Patients enrolled in the study were intended to undergo a cytoreductive nephrectomy, metastasectomy, or repeat biopsy

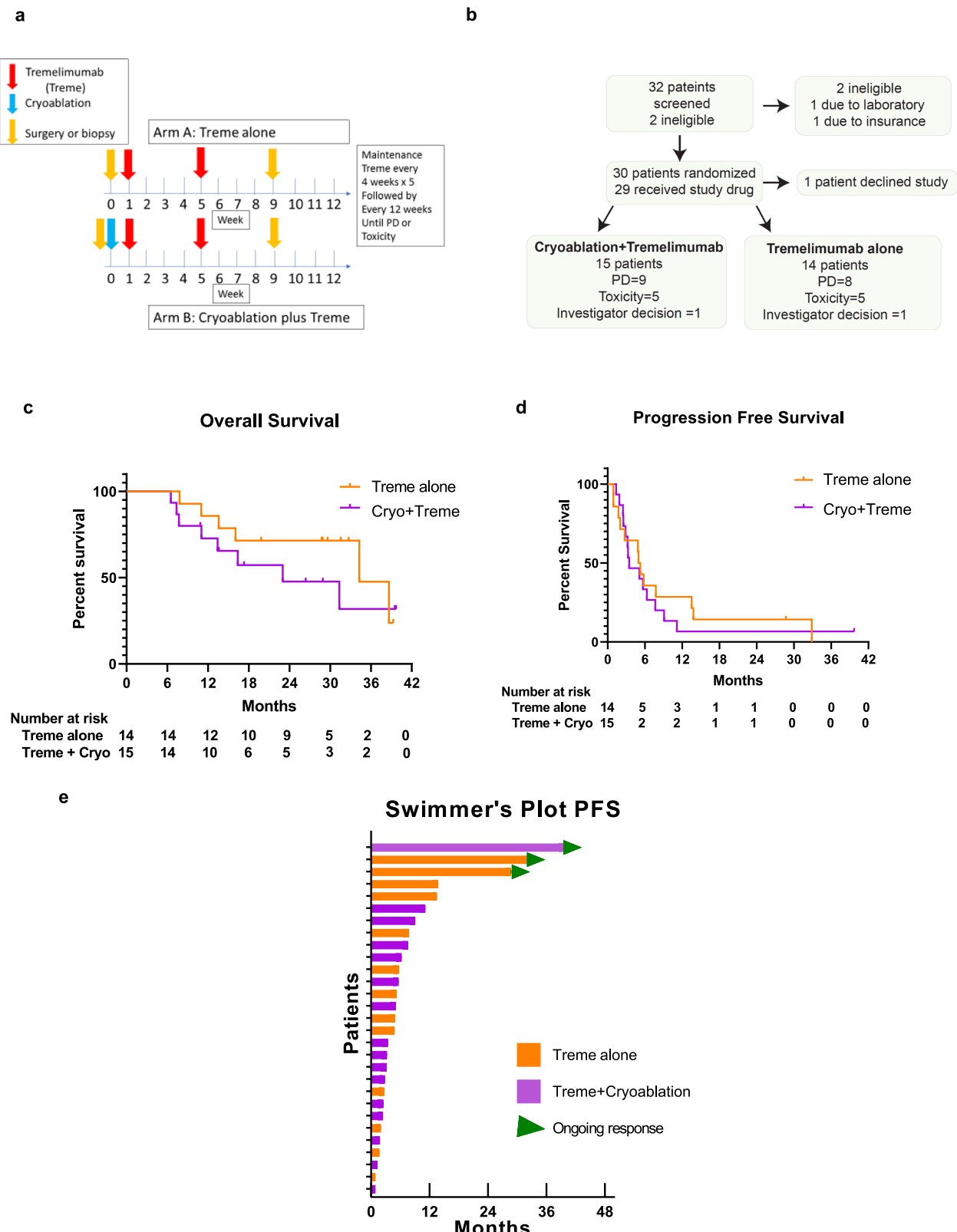

**Fig. 1 Study design and outcomes. a** Clinical trial schema. **b** Consort diagram showing patients accrued in the study. **c** and **d** Kaplan–Meier curves showing estimates of **c** overall survival and **d** progression-free survival, orange lines denote Treme alone arm and purple line denotes Cryo+Treme arm (**e**) Swimmer's Plot with PFS based on treatment arm. Orange lines denote patients in the Treme alone arm, purple lines denote patients in the Cryo+Treme arm and green triangles show patients with an ongoing response. *Treme* tremelimumab monotherapy, *Cryo+Treme* Cryoablation+Tremelimumab combination therapy, *PD* progressive disease, *PFS* progression-free survival.

**Table 1 Patient characteristics of the study.**

| Patient characteristics | | All N (%) | Treme alone N (%) | Cryo+Treme N (%) |
|---|---|---|---|---|
| All | | 29 (100%) | 14 (100%) | 15 (100%) |
| Age–median (min, max) | N = 29 | 59 (23, 82) | 64 (33, 82) | 58 (23, 80) |
| Gender | Female | 5 (17%) | 2 (14%) | 3 (20%) |
| | Male | 24 (83%) | 12 (86%) | 12 (80%) |
| Race/ethnicity | Hispanic or Latino | 3 (10%) | 3 (21%) | 0 (0%) |
| | White or Caucasian | 26 (90%) | 11 (79%) | 15 (100%) |
| Surgery type (stratification factor) | | | | |
| | Nephrectomy | 19 (66%) | 9 (64%) | 10 (67%) |
| | Metastasectomy | 3 (10%) | 2 (14%) | 1 (7%) |
| | Biopsy | 7 (24%) | 3 (21%) | 4 (27%) |
| ECOG | | | | |
| | 0 | 18 (62%) | 8 (57%) | 10 (67%) |
| | 1 | 9 (31%) | 5 (36%) | 4 (27%) |
| | 2 | 2 (7%) | 1 (7%) | 1 (7%) |
| Histology | | | | |
| | Clear | 18 (62%) | 9 (64%) | 9 (60%) |
| | Papillary | 3 (10%) | 3 (21%) | 0 (0%) |
| | Sarcomatoid | 1 (3%) | 1 (7%) | 0 (0%) |
| | Unclassified | 7 (24%) | 1 (7%) | 6 (40%) |
| Prior nephrectomy | | | | |
| | No | 17 (59%) | 7 (50%) | 10 (67%) |
| | Yes | 12 (41%) | 7 (50%) | 5 (33%) |
| IMDC score | | | | |
| | 0 | 4 (14%) | 4 (29%) | 0 (0%) |
| | 1 | 10 (34%) | 6 (43%) | 4 (27%) |
| | 2 | 10 (34%) | 3 (21%) | 7 (47%) |
| | 3 | 4 (14%) | 0 (0%) | 4 (27%) |
| | 5 | 1 (3%) | 1 (7%) | 0 (0%) |
| Number of metastatic sites–median (min, max) | N = 29 | 2 (1, 5) | 2 (1, 3) | 2 (1, 5) |
| Number of prior therapies–median (min, max) | N = 29 | 0 (0, 5) | 0 (0, 3) | 0 (0, 5) |
| Baseline measure sum–median (IQR) | N = 29 | 11.8 (7.6, 16.0) | 9.6 (4.8, 15.9) | 13.0 (9.9, 17.3) |

Baseline characteristics of study participants who received therapy on the study. Arm A tremelimumab monotherapy (Treme Alone), Arm B cryoablation–tremelimumab combination therapy (Cryo+Treme), IMDC–International Metastatic Renal Cell Carcinoma Database Consortium.

after two doses of tremelimumab at the discretion of the treating medical and urologic oncologist. Of the patients enrolled, 11 had previously undergone a radical nephrectomy and one had undergone a previous partial nephrectomy. In total, 11 of 29 patients underwent surgery on the study. In the tremelimumab monotherapy arm, three patients underwent a cytoreductive nephrectomy with retroperitoneal lymph node dissection and ipsilateral adrenalectomy. Two additional patients underwent a partial nephrectomy, and one patient underwent a metastasectomy of an adrenal metastasis. In the cryo-tremelimumab combination therapy arm, four patients underwent cytoreductive nephrectomy with retroperitoneal lymph node dissection with two patients undergoing ipsilateral adrenalectomy owing to a metastatic tumor, and one patient underwent a metastasectomy involving resection of bowel, liver, mesenteric, and peritoneal metastasis. One patient developed treatment-related colitis during the recovery phase from surgery. There were no other treatment-related toxicities associated with the surgery or tissue collection phases of the study. Supplementary Table 3 details surgical findings and Supplementary Table 4 outlines the tissue collection on the study. Ten patients did not have tissue collection after two doses of tremelimumab due to either toxicity or progressive disease on the study.

**Objective response and survival data.** The median follow-up for all patients was 29 months. The median OS for all study partici-pants was 33.7 months (95% CI: 16 months, not reached [NR]).

Figure 1c shows the OS based on the treatment arm, with a median OS of 22.7 months (95% CI: 7.8, NR months) for patients who received cryo-tremelimumab combination therapy and 33.7 months (95% CI: 13.0, NR months) for patients who received tremelimumab monotherapy. The hazard ratio for cryo-tremelimumab vs tremelimumab was 1.68 (95% CI: 0.58, 4.87). The median progression-free survival (PFS) for all treated patients was 3.3 months (95% CI: 2.0, 5.3 months). At 6 months, 24.1%, and at 12 months, 20.7% of study participants remained free from disease progression. Figure 1d shows the PFS based on the treatment arm, which was 3.0 months (95% CI: 1.7, 3.4 months) for patients who received cryoablation plus tremelimumab and 5.0 months (95% CI, 2.0, 12.9 months) for patients who received tremelimumab-only. The hazard ratio for cryoablation plus tremelimumab vs tremelimumab was 1.54 (95% CI: 0.70, 3.37).

One objective response (3.4%) using RECIST v1.1 was found in the study occurring in the cryo-tremelimumab combination therapy arm. A post hoc analysis using irRECIST did not reveal additional responses[17]. Figure 1e shows a swimmer's plot of PFS based on the treatment arm. Of the 29 patients, 3 patients (patient #2, #3, #19) have stopped therapy without requiring additional systemic therapy for their mRCC, two patients received tremelimumab monotherapy and one received cryo-tremelimumab combination therapy.

The first patient (patient #2) had clear cell histology with the poor-risk disease, IMDC score of 3, experienced a partial remission on the cryo-tremelimumab combination therapy arm and due to known bone metastasis was not labeled as a complete response. The patient has been off all systemic treatment for

**Table 2 Treatment-related grade 3 or higher toxicity.**

| Adverse event | Treme alone (N = 14) | | Cryo+Treme (N = 15) | | Total (N = 29) | |
|---|---|---|---|---|---|---|
| | **All** N (%) | **Grade 3+** N (%) | **All** N (%) | **Grade 3+** N (%) | **All** N (%) | **Grade 3+** N (%) |
| Any event | 14 (100%) | 6 (43%) | 15 (100%) | 10 (67%) | 29 (100%) | 16 (53%) |
| Laboratory | 14 (100%) | 2 (14%) | 12 (80%) | 6 (40%) | 26 (87%) | 8 (27%) |
| Colitis | 2 (14%) | 2 (14%) | 3 (20%) | 3 (20%) | 5 (17%) | 5 (17%) |
| Rash | 10 (71%) | 1 (7%) | 11 (73%) | 2 (13%) | 21 (70%) | 3 (10%) |
| Diarrhea | 1 (7%) | 0 (0%) | 6 (40%) | 3 (20%) | 7 (23%) | 3 (10%) |
| Autoimmune disorder | 0 (0%) | 0 (0%) | 2 (13%) | 2 (13%) | 2 (7%) | 2 (7%) |
| Fatigue | 5 (36%) | 1 (7%) | 9 (60%) | 0 (0%) | 14 (47%) | 1 (3%) |
| Pruritus | 6 (43%) | 0 (0%) | 7 (47%) | 1 (7%) | 13 (43%) | 1 (3%) |
| Nausea | 2 (14%) | 0 (0%) | 5 (33%) | 1 (7%) | 7 (23%) | 1 (3%) |
| Insomnia | 2 (14%) | 0 (0%) | 3 (20%) | 1 (7%) | 5 (17%) | 1 (3%) |
| Abdominal symptom | 1 (7%) | 0 (0%) | 2 (13%) | 1 (7%) | 3 (10%) | 1 (3%) |
| Fever | 2 (14%) | 0 (0%) | 1 (7%) | 1 (7%) | 3 (10%) | 1 (3%) |
| Vomiting | 1 (7%) | 0 (0%) | 2 (13%) | 1 (7%) | 3 (10%) | 1 (3%) |
| Edema | 0 (0%) | 0 (0%) | 2 (13%) | 1 (7%) | 2 (7%) | 1 (3%) |
| Hyperthyroidism | 1 (7%) | 0 (0%) | 1 (7%) | 1 (7%) | 2 (7%) | 1 (3%) |
| Acute kidney injury | 0 (0%) | 0 (0%) | 1 (7%) | 1 (7%) | 1 (3%) | 1 (3%) |
| Autoimmune hepatitis | 1 (7%) | 1 (7%) | 0 (0%) | 0 (0%) | 1 (3%) | 1 (3%) |
| Decreased activity | 1 (7%) | 1 (7%) | 0 (0%) | 0 (0%) | 1 (3%) | 1 (3%) |
| Decreased appetite | 1 (7%) | 1 (7%) | 0 (0%) | 0 (0%) | 1 (3%) | 1 (3%) |
| Hypophysitis | 1 (7%) | 1 (7%) | 0 (0%) | 0 (0%) | 1 (3%) | 1 (3%) |
| Pleuritic pain | 0 (0%) | 0 (0%) | 1 (7%) | 1 (7%) | 1 (3%) | 1 (3%) |
| Pneumonitis | 0 (0%) | 0 (0%) | 1 (7%) | 1 (7%) | 1 (3%) | 1 (3%) |
| Renal insufficiency | 0 (0%) | 0 (0%) | 1 (7%) | 1 (7%) | 1 (3%) | 1 (3%) |

Monotherapy (Treme alone), cryoablation–tremelimumab combination therapy (Cryo+Treme).
The number of patients with Grade 3 or higher treatment-related adverse events by treatment arm using the National Cancer Institute Common Terminology Criteria for Adverse Events (CTCAE) version 4.0. Tremelimumab monotherapy (Treme Alone), cryoablation–tremelimumab combination therapy (Cryo+Treme).
Laboratory events include: alkaline phosphatase increased, blood bilirubin increased, creatinine increased, hypercalcemia, hyperglycemia, hyperkalemia, hypoalbuminemia, hypocalcemia, hypokalemia, hypomagnesemia, hyponatremia, hypophosphatemia, hemoglobin a1c elevated, adrenocorticotropic hormone decreased, adrenocorticotropic hormone increased, neutrophilia albumin elevated, alkaline phosphatase increased (bone), amylase decreased, bun elevated, free T4 decreased, lactate dehydrogenase elevated, hyperphosphatemia, thrombocytosis, erythrocyte sedimentation rate increased, thyroid-stimulating hormone elevated, total protein elevated, leukocytosis, thyroid peroxidase antibody increased, lipase increased, ionized calcium decrease, proteinuria, and serum amylase increased.

>15 months at the time of the last follow-up. The second patient (patient #3) with durable benefit had clear cell histology, good risk disease, and was treatment-naive. After treatment with tremelimumab monotherapy for greater than two years, the patient developed slight growth of his oligometastatic pulmonary nodule and underwent surgical consolidation. Since that time the patient has remained off all therapy for greater than 10 months at the time of the last follow-up. The final patient (patient #19) had clear cell histology, good risk disease, and had previously received treatment with pazopanib followed by nivolumab. This individual developed immune-related hepatitis after two doses of tremelimumab monotherapy and has been off all treatment for the last 30 months with evidence of continued treatment response. The patient's baseline CT scan of her chest and her CT scan of her chest 18 months later is shown in Fig. 2a, demonstrating evidence of prolonged treatment response of her metastatic pulmonary disease. The post-treatment tumor tissues of this patient showed an increased infiltration with T cells (CD3+, CD8+), CD20+ B cells, and immune cells expressing Granzyme B and PD-1 (Fig. 2b). Of note, all three patients with durable treatment benefit of greater than 2 years had clear cell histology (swimmer's plot Fig. 2c), which led our group to perform a post hoc analysis on treatment-related outcomes as well as tissue correlates for clear cell and non-clear cell participants on the study.

**Clear cell vs non-clear cell histology outcomes.** As the study was histology agnostic, 18 mRCC patients with clear cell (mccRCC) and 11 mRCC patients with non-clear (mnccRCC) histologies were enrolled and received at least one dose of tremelimumab. The median PFS was 4.3 (95% CI: 2.0, 10.2 months) versus 3.0 (95% CI: 1.7–4.8 months) in favor of patients with clear cell

histology with a hazard ratio of 0.5 (95% CI: 0.22,1.13). Patients with mccRCC had improved outcomes as compared to those with mnccRCC regardless of treatment arm. The median OS was 33.7 (95% CI 16 months, NR) versus 16.2 months (95% CI 7.8, 33.7 months) in favor of patients with mccRCC with a hazard ratio of 0.3 (95% CI: 0.10,0.91) as shown in Fig. 2c.

Given that all three patients with prolonged responses had mccRCC we sought to compare patients with mccRCC and mnccRCC for our tissue correlative studies. For the 29 participants treated on the study, 26 patients had baseline tissue collected: 15 with mccRCC histology and 11 with mnccRCC histology, and 20 patients had additional tissue collected after two doses of tremelimumab: 12 with mccRCC histology, and 8 with mnccRCC histology (Supplementary Table 3). Supplementary Table 4 lists the samples used for immune monitoring studies.

**Association of inflamed and favorable tumor immune microenvironment with clear cell histology.** In pre-treatment tissue samples from mRCC patients with mccRCC, we observed an inflamed tumor microenvironment with enriched expression of genes belonging to both the vascular endothelial growth factor (VEGF) and interferon gamma (IFN-γ) signaling pathways (Fig. 3a, b and Supplementary Fig. 1a, b). In line with a favorable immune microenvironment, the pre-treatment tumor tissue samples of patients with mccRCC showed a preferential enrichment of Type 1 helper T (Th1) cells (Fig. 3c). Pre-treatment tissue samples from mnccRCC patients on the other hand showed an enrichment of genes belonging to unfavorable and tumor promoting neutrophil degranulation pathway (Fig. 3b and Supplementary Fig. 1c). In addition, treatment with tremelimumab monotherapy or cryo-tremelimumab combination therapy led to

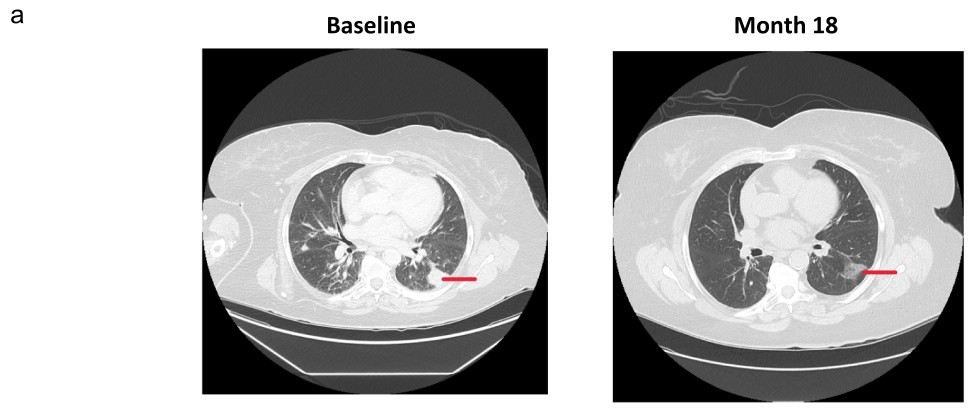

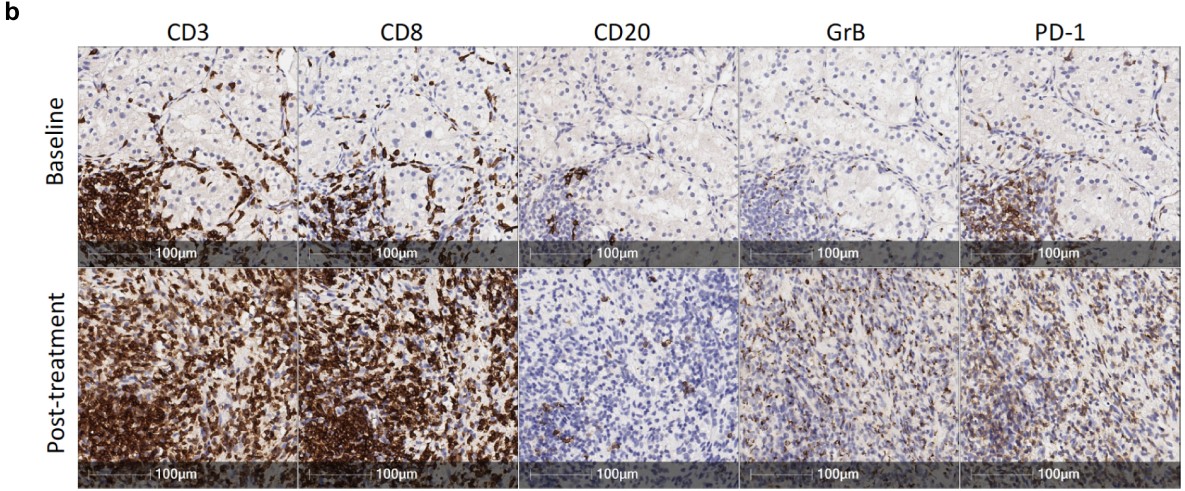

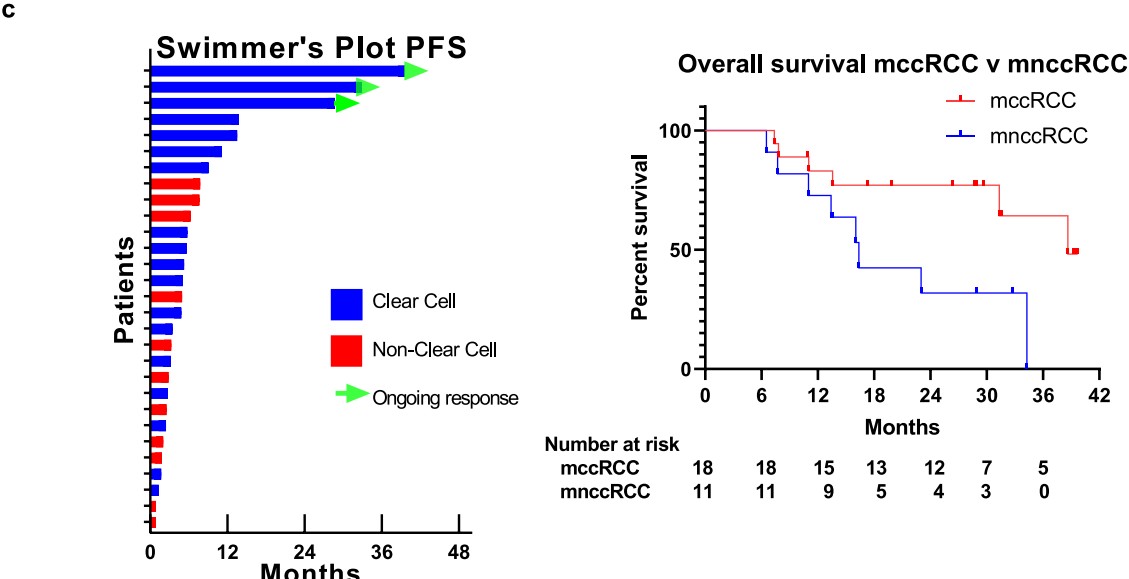

favorable immunological changes in samples from patients with mccRCC. Gene expression analysis showed an increase in CD8 T cells ($p = 0.0008$; adj $p = 0.0024$), B cells ($p = 0.04$; adj $p = 0.04$) and cytotoxic cells ($p = 0.03$; adj $p = 0.04$) in post-treatment tissues of patients with CC histology (Fig. 4a). Similarly, IHC analysis of pre and post-treatment tissues showed a significant increase in percentage of T cells (CD3+ ($p = 0.003$; adj $p = 0.015$) and CD8+

($p = 0.03$; adj $p = 0.03$), CD20+ B cells ($p = 0.02$; adj $p = 0.03$) and immune cells expressing Granzyme B ($p = 0.03$; adj $p = 0.03$) and programmed death receptor 1 (PD-1) ($p = 0.03$; adj $p = 0.03$) in post-treatment tissues of patients with mccRCC (Fig. 4b).

Recent studies have demonstrated a correlation of TLS with response to ICT in urothelial carcinoma, RCC, melanoma, and sarcoma[18–21]. Since we observed an increase in B- and T-cell

**Fig. 2 Durable responses in mRCC patients with clear cell histology. a** Patient #19 with clear cell histology with an ongoing radiographic response showing baseline cross-sectional imaging (2.5 mm cuts) showing lung metastasis (31.5 mm × 27.5 mm) and post-treatment response at 18 months (marked with red line). Radiographic analysis related to the image shown was performed once. **b** Patient #19 with favorable changes in the tumor immune microenvironment. Single stain IHC images showing an increased infiltration with T cells, B cells, and immune cells expressing Granzyme B (GrB) and PD-1 post treatment with two doses of tremelimumab. The analysis shown in the micrographs was performed once. **c** Swimmer's plot with PFS based on histology. Kaplan–Meier curve of overall survival in mccRCC versus mnccRCC patients. Blue lines denote mccRCC patients, red lines denote mnccRCC patients, and green arrows in the swimmer's plot show patients with ongoing responses. *IHC* immunohistochemistry, *PFS* progression-free survival, *mccRCC* metastatic clear cell renal cell carcinoma, *mnccRCC* metastatic non-clear cell renal cell carcinoma.

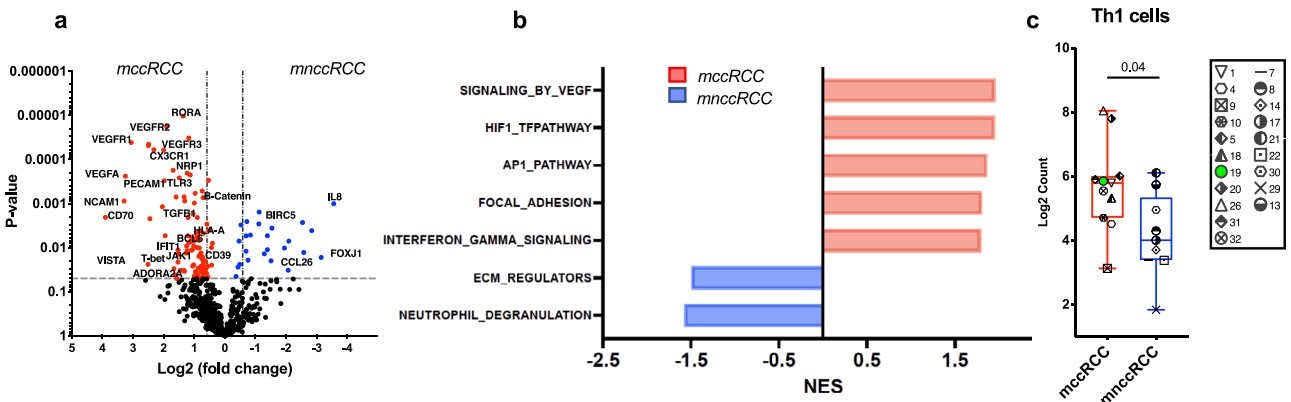

**Fig. 3 Patients with mccRCC have a favorable tumor immune microenvironment. a–c** Nanostring analysis comparing pre-treatment tissue samples of 20 mRCC patients with clear cell (mccRCC) versus non-clear cell (mnccRCC) histology (mccRCC, $n = 11$; mnccRCC, $n = 9$). **a** Volcano plot illustrating differentially expressed genes (DEGs). Colored points above the cutoff in the plot show DEGs with $p < 0.05$ and fold change ≥1.5 that are either upregulated (red) or downregulated (blue) in mccRCC compared to mnccRCC samples. Two-sided *P* values are from Welch's *t* test. **b** Bar plots showing differentially activated pathways in tumor tissue samples from patients with mccRCC versus mnccRCC. Upregulated pathways in mccRCC are indicated by red bars while downregulated pathways are indicated by blue bars. *P* values (FDR-adjusted <0.2) are from GSEA algorithm and **c** Boxplot showing infiltration with Th1 cells. The ends of the box are the upper and lower quartiles (75th and 25th percentiles), the median is the horizontal line inside the box, and whisker lines extend to outliers (minimum and maximum). *P* value was calculated using the two-tailed unpaired Mann–Whitney test. Each patient is identified by a unique symbol as shown in the inset on the right. Red box whiskers show mccRCC and blue box whiskers show mnccRCC cohort of patients. Patients with ongoing responses are shown in green. $P < 0.05$ denotes significant differences. *mccRCC* metastatic clear cell renal cell carcinoma, *mnccRCC* metastatic non-clear cell renal cell carcinoma, *NES* normalized enrichment score, *Th1* T helper cells type 1. Source data are provided as Source Data file.

infiltration in post-treatment tissues in patients with clear cell histology (Fig. 4a, b), we assessed TLS gene signature using NanoString and the presence of TLS using IHC in the tissues of patients with mccRCC and mnccRCC. NanoString analysis showed that in pre-treatment tissues there were no differences in TLS signatures between the two groups (Fig. 4c and Supplementary Fig. 2). Treatment with tremelimumab mono-therapy or cryo-tremelimumab combination therapy led to an increase in TLS gene signature in post-treatment tissues of patients with mccRCC (Fig. 4c). IHC analysis showed a similar increase in the density of TLS in post-treatment tissues of mccRCC patients (Fig. 4d). A representative TLS showing single stain IHC images from a post-treatment tissue sample of patient#2 who had a prolonged response is shown in Fig. 4e. In summary, pre-treatment tumor tissues of mccRCC patients have higher expression of IFN-γ signaling and VEGF signaling pathways. Treatment with tremelimumab monotherapy or cryo-tremelimumab combination therapy led to an increase in immune cell infiltration and an increase in TLS in post-treatment tissues of mccRCC patients.

**Cryoablation leads to mobilization of immune cells into the tumor microenvironment of mRCC patients with clear cell histology.** Based on pre-clinical studies, we hypothesized that the addition of cryoablation would lead to an increase in immune cell infiltration within the tumor microenvironment of mRCC patients[16]. However, we did not observe significant differences between the treatment arms of tremelimumab monotherapy as compared with

cryo-tremelimumab combination therapy, which may be owing to the inherent heterogeneity of our study population consisting of both mccRCC and mnccRCC as well as the small number of patients available for analysis. In a post hoc analysis, we focused our studies on a comparison between mccRCC and mnccRCC and found that patients with mccRCC had favorable changes in the immune microenvironment and better clinical outcomes. In addition, we compared immune responses in mccRCC tumor samples after treatment with tremelimumab monotherapy as compared with cryo-tremelimumab combination therapy. We observed a significant increase in T cells (CD3+ ($p = 0.002$; adj $p = 0.008$) and CD8+ ($p = 0.009$; adj $p = 0.018$) in the total tumor microenvironment upon treatment with cryo-tremelimumab combination therapy compared with tremelimumab monotherapy (Fig. 5a). The effect of cryoablation was more apparent when we dissected the total tumor micro-environment into the tumor and stromal regions and analyzed the distribution of immune cells in these two regions. There was a significant increase in T cells (CD3+ and CD8+) in both the tumor (CD3+, $p = 0,009$; adj $p = 0.036$, CD8 +, $p = 0.04$; adj $p = 0.04$) and stromal regions (CD3+, $p = 0.001$; adj $p = 0.004$, CD8 +, $p = 0.009$; adj $p = 0.018$) within the TME after treatment with cryo-tremelimumab combination therapy. Although CD20+ B cells ($p = 0.06$; adj $p = 0.08$) and PD-1+ immune cells ($p = 0.13$; adj $p = 0.13$) were not significantly different in the total tumor micro-environment and stromal areas for the two treatment arms (Fig. 5a), we noted that there was a significant increase in both these cell types, specifically within the tumoral areas in post-treatment tumor tissue samples of patients treated with cryo-tremelimumab combination

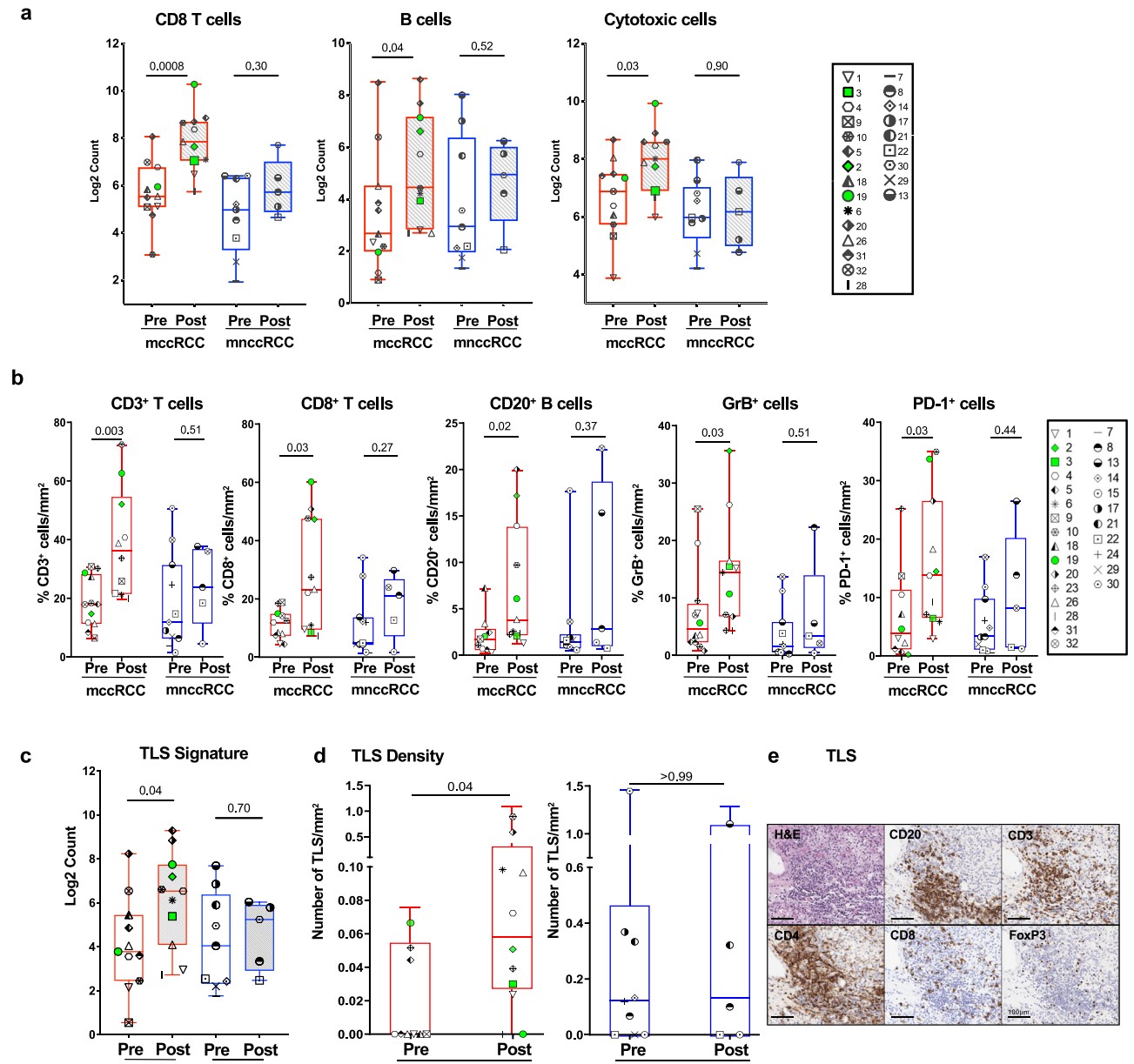

**Fig. 4 Treatment with tremelimumab monotherapy or cryo-tremelimumab combination therapy leads to favorable changes in post-treatment tissue samples of patients with clear cell histology. a–d** Pre and post-treatment tissue samples of patients with clear cell (mccRCC) and non-clear cell (mnccRCC) histology were analyzed by NanoString (**a** and **c**, mccRCC: pre ($n = 11$), post ($n = 11$); mnccRCC: pre ($n = 9$), post ($n = 5$)) and IHC (**b**, **d** and **e**, mccRCC: pre ($n = 13$), post ($n = 11$); mnccRCC: pre ($n = 11$), post ($n = 5$)). Box plots showing **a** infiltration of CD8 T cells, B cells, and cytotoxic cells, **b** percentage of CD3 T cells, CD8 T cells, CD20+ B cells, and immune cells expressing Granzyme B and PD-1, **c** TLS gene signature score and, **d** TLS density. For box plots (**a–d**), the ends of the boxes are the upper and lower quartiles (75th and 25th percentiles), the median is the horizontal line inside the box, and whisker lines extend to outliers (minimum and maximum). Each patient is identified by a unique symbol as shown in the inset on the right. Patients with ongoing responses are shown in green. Red box whiskers show mccRCC and blue box whiskers show mnccRCC cohort of patients. P values were calculated using the two-tailed unpaired Mann–Whitney test. $P < 0.05$ denotes significant differences. P values were adjusted for multiple comparisons using the Benjamini–Hochberg procedure. **e** Representative single stain IHC images showing a TLS from a post-treatment tumor tissue sample of patient #2 with clear cell histology who has an ongoing response. The analysis shown in the micrographs was performed once. *IHC* immunohistochemistry, mccRCC; metastatic clear cell renal cell carcinoma, *mnccRCC* metastatic non-clear cell renal cell carcinoma, *Pre* pre-treatment sample, *Post* post-treatment sample, *GrB* granzyme B, *TLS* tertiary lymphoid structures. All source data are provided as a Source Data file.

therapy as compared with tremelimumab monotherapy (CD20+ B cells–tumor area, ($p = 0.03$; adj $p = 0.04$), stromal area ($p = 0.11$; adj $p = 0.14$); PD-1+ immune cells–tumor area, ($p = 0.03$; adj $p = 0.04$), stromal area ($p = 0.25$; adj $p = 0.25$) (Fig. 5b). In summary, our findings in this small study indicate that combination therapy with tremelimumab plus cryoablation may lead to favorable immunologic

changes in the tumor microenvironment of mccRCC patients, which will require validation in future trials with larger cohorts of patients.

## Discussion

Treatment with tremelimumab with or without cryoablation was associated with immune-mediated toxicities with grade 3 or

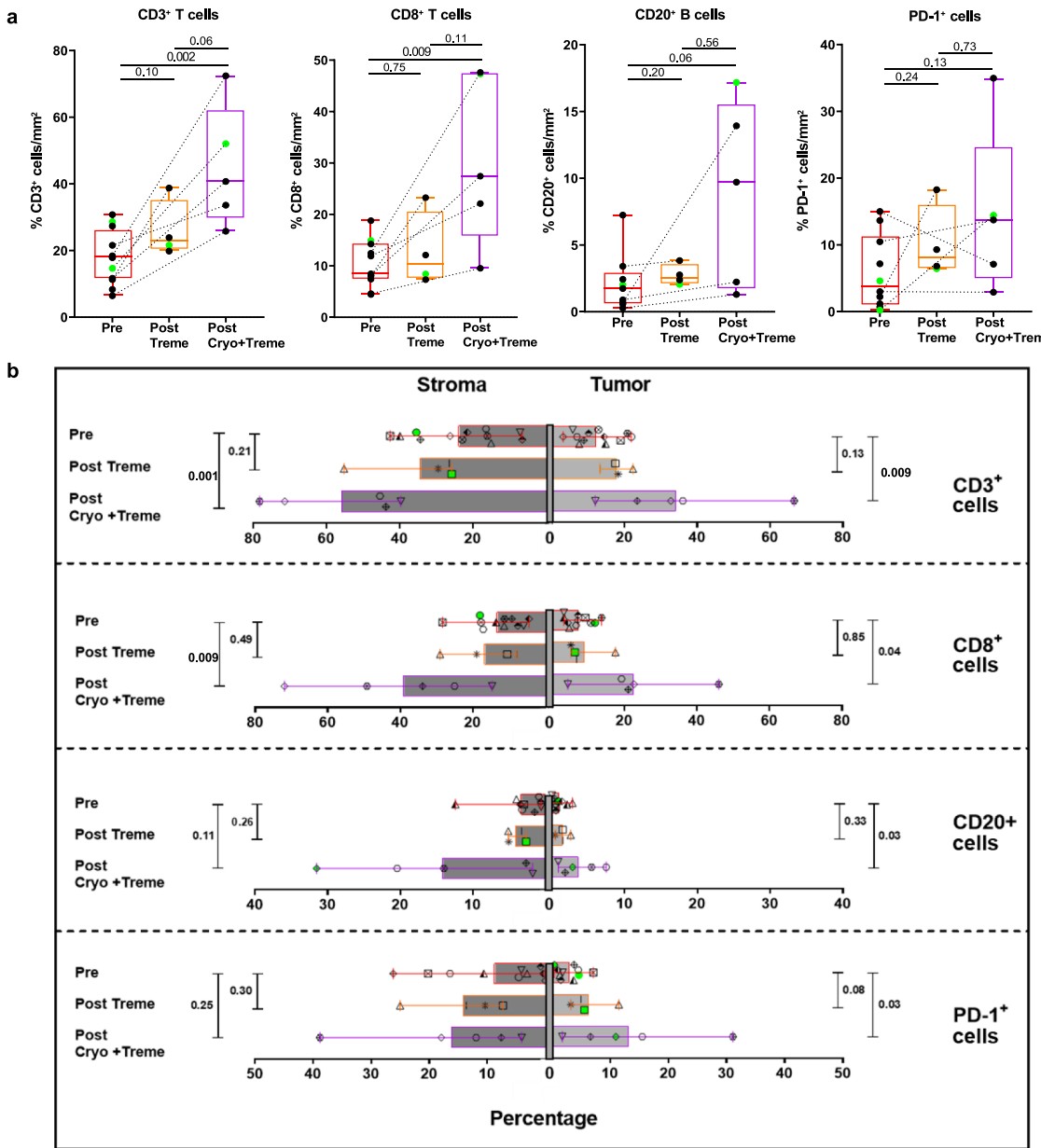

**Fig. 5 Adding cryoablation to tremelimumab leads to immune cell infiltration in the tumor microenvironment of mRCC patients with clear cell histology.** Pre- and post-treatment tissue samples from patients with clear cell histology who had evaluable tumor tissue samples were analyzed for immune cell infiltration by IHC. **a** Box plots showing infiltration of CD3 and CD8 T cells, CD20 + B cells, and immune cells expressing PD-1 in the tumor microenvironment of pre, (red, n = 12) and post-treatment tumor samples of patients treated with tremelimumab monotherapy (post Treme, orange, n = 4) or cryoablation plus tremelimumab combination therapy (Post Cryo+Treme, purple, n = 5). For the box plots, the ends of the box are the upper and lower quartiles (75th and 25th percentiles), the median is the horizontal line inside the box, whisker lines extend to outliers (minimum and maximum) and **b** Bar plots showing the preferential distribution of immune cells as in **a** between the tumor and stromal compartments of the tumor microenvironment. Pre shown in red, post Treme shown in orange, and Post Cryo+Treme shown in purple. Patients with ongoing responses are shown in green. Error bars represent the mean with minimum to maximum value range. P values were calculated using a two-tailed, unpaired Mann–Whitney test. P values were adjusted for multiple comparisons using the Benjamini–Hochberg procedure. P < 0.05 denotes significant differences. *Pre* pre-treatment sample, *Post Treme*, post-treatment sample treated with tremelimumab monotherapy, *Post Cryo+Treme* post-treatment sample treated with cryoablation plus tremelimumab combination therapy. All source data are provided as a Source Data file.

greater events occurring in 55% of patients. These rates of grade 3 or greater adverse events are similar to combination studies with ICT in RCC and in studies with ipilimumab alone or in combination with nivolumab in RCC when the dose of ipilimumab was 3 mg/kg IV every 3 weeks[11,22–26]. The immune-mediated toxicities improved after treatment with appropriate immunosuppressive agents. The toxicity encountered in the study was likely

related to the dosing of tremelimumab at 10 mg/kg, which results in relatively larger doses for patients weighing more than 75 kg, as compared to more contemporary studies with tremelimumab where the dose is capped. The addition of cryo to a metastatic site did not increase toxicity. Of note, treatment with tremelimumab with or without cryoablation led to a low frequency of responses, but when responses occurred, they were durable in a subset of

patients with clear cell histology. Given the small size of the study, no direct comparisons of clinical activity were intended between the two arms.

From an immune monitoring perspective, a subset analysis in patients with mccRCC histology identified a favorable immune response within the tumor microenvironment when cryoablation was provided immediately prior to the initiation of tremelimumab. We found an inflamed and favorable immune microenvironment in pre-treatment samples and an increase in immune cell infiltration in post-treatment samples obtained from mccRCC patients. In agreement with previous studies in clear cell RCC, we observed higher expression of IFN-gamma and VEGF signaling pathways in pre-treatment tissues of mccRCC patients as compared to mnccRCC[27]. A favorable immunogenic TME with high expression of the IFN-gamma pathway has been shown to predict clinical responses to ICT[28]. Th1 gene expression has also been associated with clinical responses and our analysis demonstrated a higher Th1 signature in pre-treatment tissues of mccRCC compared to mnccRCC[29]. In addition, treatment with anti-CTLA-4 led to favorable immunologic changes in the TME of mccRCC cases which is concordant with published studies[26,30–32].

In addition, our group and others have shown a positive association between the presence of TLS and response to ICT. Our previous studies in mccRCC patients showed a positive correlation with TLS and response to ICT consisting of nivolumab monotherapy or nivolumab combined with bevacizumab or ipilimumab. Here, we are reporting an increase in TLS density upon treatment with anti-CTLA-4 antibody (tremelimumab) and this increase is specific to patients with mccRCC. Our findings show that both tremelimumab monotherapy and cryotremelimumab combination therapy can lead to an increase in TLS in mRCC patients with clear cell histology.

In support of our hypothesis that cryoablation would lead to a detectable increase in immune cell infiltration within the tumor microenvironment, we detected favorable immune modulation post cryoablation in mccRCC patients. Necrotic cell death caused by cryoablation releases intracellular danger signals, which can be recognized by the immune system and can potentially initiate or augment a tumor-specific immune response. Necrotic cell death, for example, has been shown to release heat shock proteins, which leads to activation of antigen-presenting cells[33]. A favorable immune cell infiltration in response to cryoablation was observed in an orthotopic renal cell carcinoma murine model[15]. In humans, cryoablation has been shown to induce an immune response and an increase in T-cell clones in stage 1 clear cell RCC patients[34]. A pilot study evaluating cryoablation plus granulocyte macrophage-colony stimulating factor combination therapy in six mRCC patients with lung metastasis showed induction of cellular and humoral antitumor responses with induction of Th1 cytokines in blood samples from four out of the six patients enrolled in the study[35]. A previous study in breast cancer assessing preoperative cryoablation alone, ipilimumab alone and, cryoablation in combination with single-dose ipilimumab in 19 breast cancer patients showed a systemically sustained increase in activated and proliferating T cells and an increase in the ratio of proliferating T effector cells relative to regulatory T cells within the tumor microenvironment in the cryo-ipilimumab combination therapy arm[36]. Our initial analysis did not show differences between the two arms, which was likely owing to enrollment of all subtypes of RCC that are known to have different drivers of tumor development and responses to therapy. Another challenge in detecting the immune-modulatory activity resulting from cryoablation was the fact that the pre-treatment biopsy was taken from the lesion that eventually underwent cryoablation, which meant that a different lesion was used for the post-treatment biopsy, thereby

making comparisons between pre- and post-treatment samples difficult. An ideal scenario would have been to sample a second target lesion at the time of ablation and then repeat the sampling of the alternative lesion for consistency. Despite these shortcomings, in the post hoc analyses, we observed a favorable effect of cryoablation plus tremelimumab on immune responses in tumor tissues from patients with mRCC patients who had mccRCC. Tremelimumab with or without cryoablation led to immune-related toxicities that were manageable in patients with mRCC. A small subset of patients experienced durable disease control with anti-CTLA-4 monotherapy or combination therapy with anti-CTLA-4 plus cryoablation. Cryoablation prior to initiation of immune checkpoint blockade did not increase toxicity or response but led to a detectable increase in immune cells within the tumoral microenvironment of evaluable mRCC patients with mccRCC. Patients with clear cell histology appear to have a more favorable response to anti-CTLA-4 therapy as compared with non-clear cell histologies owing to a pre-existing TME with higher IFN-γ expression, favorable increase in tumor-infiltrating immune cells and, an increase in TLS.

There is an unmet clinical need for understanding the differences between mccRCC and mnccRCC with an aim to select patients' groups for developing safe and effective treatment strategies. This study shows an increase in T-cell infiltration in the tumor microenvironment of mccRCC patients upon cryo-tremelimumab combination therapy and provides the rationale for future clinical trials focusing on mccRCC patients.

## Methods

**Study and patient cohorts.** NCT02626130 was approved by The University of Texas MD Anderson Cancer Center's IND office with oversight by our Institutional Review Board (IRB), Immunotherapy Platform umbrella protocol PA13-0291. The study was compliant with all relevant regulations pertaining to the use of human study participants and was conducted in accordance with the criteria set by the Declaration of Helsinki. This was a pilot study of patients with mRCC randomized 1:1 to receive tremelimumab (anti-CTLA-4) alone or after cryoablation of a metastatic lesion. Cryoablation was performed using the standard of care cryoablation systems [Endocare; Varian Systems, USA] under computed tomography guidance by interventional radiologists with at least 5 years of experience in these procedures. Metastatic lesions contributing to cancer-related symptoms (i.e., pain) were selected. Metastatic lesions in the soft tissues, visceral organs, or bone were considered eligible for ablation. Cryoablation margins were kept at least 1 cm away from critical structures (i.e., blood vessels, nerves, bowel loops) to prevent thermal injury.

All patients provided informed consent on an Institutional Review Board-approved clinical protocol (NCT02626130) to participate in the study. Key inclusion criteria included an allowance for any renal cell histology, the requirement of a cryoablation eligible lesion as per our interventional radiology collaborators, measurable disease as defined by RECIST v1.1 unless that patient was eligible for cytoreductive nephrectomy, an ECOG performance status less than or equal to 3. Key exclusion criteria involved a history of immune-related disorders except for Hashimoto's thyroiditis, type I diabetes mellitus, or psoriasis not requiring topical corticosteroids, previous therapy with an anti-CTLA-4 agent, and untreated brain metastasis. For a full list, please refer to the supplementary note 1.

The primary endpoint of the study was safety using the National Cancer Institute (NCI) Common Terminology Criteria for Adverse Events (CTCAE) version 4.0. Each arm was followed for the rate of extreme toxicity, which was defined as any grade 3 or higher adverse event occurring in the first two cycles that did not improve in severity within 2 weeks of corticosteroid therapy. Safety monitoring was continuous after the 5th patient was enrolled with strict stopping rules if extreme toxicity was found at an unacceptable rate in either arm based on Bayesian rules to stop the arm if Pr(θT > 0.25 | data) >0.85 where θT is the proportion of patients with extreme toxicity, assuming a prior of beta(0.5, 1.5)[37]. Based on the stopping rules, with 15 patients in each arm, either arm was designed to stop early 19% of the time if the true toxicity rate is 25%, only 1% if the rate is as low as 10%, and 58% if it is 40%. Secondary endpoints included overall response rate using Response Evaluation Criteria in Solid Tumors v1.1 (RECIST v1.1), PFS defined as the time from study enrollment until disease progression or death, and OS defined as the time from study enrollment until disease progression or death[38]. The Kaplan–Meier method was used for both progression-free survival and OS with hazard ratios and the 95% confidence intervals calculated using the log-rank test. The hazard ratio and 95% confidence interval were computed using Cox regression [SAS v9.4, The SAS Institute Inc., Cary, NC, USA]. The dose of tremelimumab was 10 mg/kg intravenous infusion every 4 weeks for two doses

followed by tissue collection with the resumption of tremelimumab every 4 weeks for three additional doses followed by every 12-week administration. Correlative studies included planned immune monitoring studies. Post hoc analysis included a comparison of patients with clear cell and non-clear cell histologies. The first patient was enrolled on 11/7/2016 and the final patient was enrolled on 10/25/2018.

The immunological studies were performed on baseline biopsy samples, which could be from the primary tumor or a metastatic site, and then from either a surgical specimen or repeat biopsy after two doses of tremelimumab. Pre- and post-treatment tissue samples were collected for correlative analyses after patients provided informed consent on an Institutional Review Board-approved laboratory protocol (PA13-0291).

**Nanostring**. RNA was isolated from formalin-fixed paraffin-embedded (FFPE) tumor sections by de-waxing using deparaffinization solution (Qiagen, Valencia, CA), and total RNA was extracted using the RecoverALL™ Total Nucleic Acid Isolation kit (Ambion, Austin, TX) according to the manufacturer's instructions. The RNA purity and quantity were assessed on the ND-Nanodrop1000 spectrometer (Thermo Scientific, Wilmington, MA, USA). For the NanoString assay, 100 ng of RNA was used to detect immune gene expression using nCounter PanCancer Immune Profiling panel along with a custom CodeSet. Counts of the reporter probes were tabulated for each sample by the nCounter Digital Analyzer and raw data output was imported into nSolver version 4.0 (http://www.nanostring.com/products/nSolver). nSolver (advanced analysis 2.0) data analysis package was used for normalization, cell type analysis, and differential gene expression. Hierarchical clustering heatmap analysis and Gene Set Enrichment Analysis (GSEA) were performed with Qlucore Omics Explorer version 3.7 software (Qlucore, NY, USA). Data were plotted using GraphPad Prism 8 (GraphPad Software v-8.4.3) and two-tailed Mann–Whitney tests were performed to compare two groups, $p$ value <0.05 were considered significant. $P$ values were adjusted for multiple comparisons using the Benjamini–Hochberg procedure. Welch's $t$ test (determined by nSolver advanced analysis 2.0 software) was used to determine $p$ values for volcano plot (differential expression) and $p$ value (FDR-adjusted<0.2) was used for GSEA analysis.

**Immunohistochemistry analysis**. Hematoxylin and eosin staining and immunohistochemistry (IHC) were performed on 4 μm FFPE pre and post-treatment tissue samples. IHC analysis on subsequent sections was performed using antibodies against CD3 (Agilent Dako, cat#A0452, 1:100), CD20 (Agilent Dako, cat#M075501-2, 1:1400), CD8 (Thermo Scientific, cat# MS-457-S, 1:25), Granzyme B (Leica Microsystems, cat# PA0291, RTU) and PD-1 (Abcam, cat# ab137132, 1:250). Sections were processed with peroxidase-conjugated avidin/biotin and 3'-3-diaminobenzidine substrates (Leica Biosystems, cat# DS9800), and the IHC slides were scanned and digitalized using the Scanscope XT system (Aperio/Leica Technologies). Single stain IHC quantification analysis was performed by the pathologist using the HALO 2.3.2089.70 software (Indica Labs). The number of marker positive cells for each analysis area were calculated and expressed as density (number of positive cells/mm²) and densities were plotted using Prism V8.4.3 (GraphPad). TLS quantification was done as previously described and the total number of TLS per mm² of tumor area was plotted[18]. Statistical analysis was done using a two-tailed, unpaired Mann–Whitney test, and $p$ values <0.05 were considered statistically significant. $P$ values were adjusted for multiple comparisons using the Benjamini–Hochberg procedure.

**Reporting summary**. Further information on research design is available in the Nature Research Reporting Summary linked to this article.

## Data availability

The authors declare that the data supporting the findings in this study are available in the manuscript and its supplementary information and source data files. The NanoString data that support the findings of this study are available as Supplementary Data 1 (normalized Log2 counts) with the manuscript. Other relevant data related to the current study will be available from the corresponding author (P.S.) based upon a reasonable academic request and will require a data access agreement with the University of Texas at MD Anderson Cancer Center and the requester as the information may include data collected under an institutional alliance clinical trial protocol. Source data are provided with this paper.

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

## Acknowledgements

We are thankful to the patients and their families who participated in this study. We thank our clinical trial, data, and regulatory research teams for their assistance. We would like to thank the strategic alliance team members at MedImmune/Astra Zeneca and MD Anderson Cancer Center, for their support and the ASCO Conquer Cancer Foundation for their support. We would like to thank Ashura Khan, Marla Polk other members of the Immunotherapy Platform at MD Anderson Cancer Center for their technical support and scientific input (Drs. Allison and Sharma are the executive director and scientific director of the Immunotherapy Platform). Support for the study was provided by MedImmune/Astrazeneca; ASCO Conquer Cancer Foundation.

## Author contributions

M.T.C. provided clinical care, collected the clinical data, supervised and participated in data analysis at all stages and wrote the manuscript; S.F.C. contributed to the development of the project, provided surgical care, supervised and participated in data analysis, reviewed the manuscript; A.L.T. contributed to the development of the project, provided interventional radiology expertise, provided data analysis, and reviewed the manuscript; R.A.H. provided interventional radiology expertise, provided data analysis, and reviewed the manuscript; K.A. provided interventional radiology expertise, provided data analysis, and reviewed the manuscript; B.S.T. designed the statistical plan, performed the statistical analyses and participated in writing the manuscript; P.R. analyzed the pathologic data, provided data analysis, and reviewed the manuscript; J.A.K. provided study design input, provided surgical expertise, reviewed the manuscript; C.G.W. provided study design input, provided surgical expertise, reviewed the manuscript; N.M.T provided study design input, provided clinical care, reviewed the manuscript; E.J. provided study design input, provided clinical care, reviewed the manuscript; J.G. provided study design input, provided clinical care, reviewed the manuscript; A.J.Z. provided study design input, provided clinical care, reviewed the manuscript; A.Y.S. provided study design input, provided clinical care, reviewed the manuscript; P.S and J.P.A. supervised the Immunotherapy Platform team and provided interpretation of the NanoString and IHC data and participated in writing the manuscript. H.C. performed the analysis of the NanoString data. A.B.E. analyzed the pathologic data S.B., S.J., and S.S.Y. participated in data interpretation and writing of the manuscript. F.D. participated in the writing and editing of the manuscript. All authors reviewed the manuscript at all stages.

## Competing interests

M.T.C. reports consulting/advisory roles for AstraZeneca, Astellas, Eisai, EMD Serono, Exelixis, Genentech, Pfizer, Seattle Genetics, Consulting: AXDev, Exelixis, Pfizer, Research grants: ApricityHealth, EMD Serono, Exelixis, Janssen, Pfizer, Non-CME education: Bristol Myers-Squibb, Merck, Roche, Pfizer. S.F.M. reports consulting/advisory roles for Johnson and Johnson and Research Funding from QED; Education—Ology education. A.L.T. reports consulting/advisory roles for Boston Scientific, Cello Therapeutics, Endocare, AstraZeneca; and Research Funding from Boston Scientific, Johnson & Johnson. R.A.S. reports consulting/advisory roles for Varian, Boston Scientific. J.A.K. reports consulting/advisory roles for Pfizer, Merck, Roche/Genentech, and Research Funding from Roche/Genentech, Mirati, Merck, Elypta; Stock Ownership: Allogene, MedTek, ROM Technologies. N.M.T. reports consulting/advisory roles for Bristol Myers-Squibb; Pfizer; Nektar Therapeutics; Exelisis, Inc, Eisai Medical Research; Eli Lilly; Oncorena; Calithera Bioscience; Surface Oncology; Novartis, Ipsen; Merck Sharp & Dohme and Research Funding from Bristol Myers-Squibb; Nektar Therapeutics; Calithera Bioscience; Arrowhead Pharmaceuticals, Inc. E.J. reports consulting/advisory roles for Eisai, Exelixis, Novartis, Merck, Pfizer, Roche. And Research support from Exelixis, Merck, Pfizer. J.G. reports consulting/advisory roles for Seattle Genetics, Jounce Therapeutics, Pfizer, Janssen, Symphogen, AstraZeneca, ARMO biosciences. A.J.Z. reports consulting/advisory roles for AstraZeneca and Bayer and Research funding to his institution from Infinity Pharma, Honoraria from Pfizer, and Janssen. A.Y.S. reports consulting/advisory roles for Ad boards, Pfizer, Exelixis, and Research Grant funding from Bristol Myers-Squibb, Eisai, EMD Serono. J.P.A. reports consulting, advisory roles, and/or stocks/ownership for Achelois, Apricity Health, BioAtla, Codiak BioSciences, Dragonfly Therapeutics, Forty-Seven Inc., Hummingbird, ImaginAb, Jounce Therapeutics, Lava Therapeutics, Lytix Biopharma, Marker Therapeutics, Polaris, BioNTech, and Adaptive Biotechnologies; and owns a patent licensed to Jounce (61/247,438; "Combination Immunotherapy for the Treatment of Cancer"). P.S. reports consulting, advisory roles, and/or stocks/ownership for Achelois, Apricity Health, BioAlta, Codiak BioSciences, Constellation, Dragonfly Therapeutics, Forty-Seven Inc., Hummingbird, ImaginAb, Jounce Therapeutics, Lava Therapeutics, Lytix Biopharma, Marker Therapeutics, Oncolytics, Infinity Pharma, BioNTech, Adaptive Biotechnologies, and Polaris; and owns a patent licensed to Jounce Therapeutics (61/247,438; "Combination Immunotherapy for the Treatment of Cancer"). The remaining authors declare no potential competing interest.
