## [Peer Review File · Nature Communications]

Reviewers' Comments:

Reviewer #1:

Remarks to the Author:

The authors report an interesting pilot study of tremelimumab +/- cryotherapy in mRCC. The data, albeit limited by small numbers and limited post-treatment biopsies, are hypothesis generating and relevant.

1. The abstract devotes a lot of text to OS results which are not interpretable in such a small study. I would devote more text to either safety and/or immune correlates.
2. Would be good to further detail what a 'cryotherapy eligible lesion' is.
3. Figure 3B is blank?

Reviewer #2:

Remarks to the Author:

This is to report the data from a pilot study with tremelimumab and randomized with (n=15) or without (n=14) cryoablation for mRCC patients. The primary objectives from the protocol is to assess the safety and tolerability, and the purpose of randomization is to identify potential toxicity difference between 2 arms. The statistical methods used is appropriate in general.

I have some major concerns:

1. The primary objective is to show the safety and purpose of randomization is to identify potential toxicity difference between 2 arms. The paper didn't include any statistical testing to compare two arms (5/15 for single tremelimumab arm and 5/14 for combined arm, which is similar). In the proposal, the stopping rule is designed at TOX rate 25% as standard therapy, but both arms are higher than this rate but the conclusion is "with expected toxicity". Needs to clarify what is the expected toxicity rate, if 25% is the expected, needs a one-arm comparison between each arm to this historical rate. Btw, it's not clear whether the stopping rule from protocol on TOX is controlled for grade 3 and above AE or not. This needs to be clarified.
2. The protocol proposed to use "A mixed model accounting for patient effects... to analyze the longitudinal data on immunological values over time". The paper didn't include this analysis
3. Though the sample size is small, OS and PFS curves showed clear separation between two arms, which is good but there is not statistical testing (log-rank test) or modeling (cox model with HR and 95% confidence interval) results.
4. The study conducted many two group comparisons, and didn't mention any multiplicity adjustment method, such as Bonferroni or FDR.

Some minor comments:

1. What is the barplot for Figure 3. B ? It didn't show.
2. Why total sample size is N=30 in Table 2 (and Supp Table 1) ? Treme Alone N=14 and Treme+Cryo N=15, it cannot be added to 30..

Response to Reviewers

Manuscript NCOMMS-21-17493: A pilot study of Tremelimumab with or without cryoablation in patients with metastatic renal cell carcinoma

REVIEWER COMMENTS

Reviewer #1 (Remarks to the Author): with expertise in RCC, immunotherapy, clinical trials.

The authors report an interesting pilot study of tremelimumab +/- cryotherapy in mRCC. The data, albeit limited by small numbers and limited post-treatment biopsies, are hypothesis generating and relevant.

1. The abstract devotes a lot of text to OS results which are not interpretable in such a small study. I would devote more text to either safety and/or immune correlates.

Authors' response: We appreciate the Reviewer's observation on devotion of lot of text to OS and we have modified the abstract as suggested.

2. Would be good to further detail what a 'cryotherapy eligible lesion' is.

Authors' response: We appreciate the Reviewer's request for further details on what a "cryotherapy eligible lesion" is. In the revised manuscript we have inserted the following statement into the methods section; "Metastatic lesions contributing to cancer related symptoms (i.e., pain) were selected. Metastatic lesions in the soft tissues, visceral organs or

bone were considered eligible for ablation. Cryoablation margins were kept at least 1 cm away from critical structures (i.e., blood vessels, nerves, bowel loops) to prevent thermal injury.”

3. Figure 3B is blank?

Authors' response: We apologize for the blank figure. We have addressed the Mac to PC issue and the figure is visible now.

Reviewer #2 (Remarks to the Author): with expertise in biostatistics, clinical trials

This is to report the data from a pilot study with tremelimumab and randomized with (n=15) or without (n=14) cryoablation for mRCC patients. The primary objectives from the protocol are to assess the safety and tolerability, and the purpose of randomization is to identify potential toxicity difference between 2 arms. The statistical methods used is appropriate in general.

I have some major concerns:

1. The primary objective is to show the safety and purpose of randomization is to identify potential toxicity difference between 2 arms. The paper didn't include any statistical testing to compare two arms (5/15 for single tremelimumab arm and 5/14 for combined arm, which is similar). In the proposal, the stopping rule is designed at TOX rate 25% as standard therapy, but both arms are higher than this rate, but the conclusion is “with expected toxicity”. Needs to clarify what is the expected toxicity rate, if 25% is the expected, needs a one-arm comparison between each arm to this historical rate. Btw, it's not clear whether the stopping rule from protocol on TOX is controlled for grade 3 and above AE or not. This needs to be clarified.

Authors' response: The reviewer is correct that the primary objective was safety. The purpose of the randomization was to make sure our estimates were as unbiased as possible. With only 15 patients planned per group, no formal comparisons between the two arms were planned, and as such, no statistical testing between the arms was performed, as per the protocol. The stopping rule was designed at an extreme toxicity rate of 25% but we defined extreme toxicities as toxicities that were durable (high grade related toxicities that lasted at least 2 weeks despite high dose corticosteroids) and occurred within the first 2 cycles of treatment only. Based on this definition and the stopping rules, we did not stop the trial early. Additional information has been added to the methods to clarify the details of the Bayesian stopping rules with the reference. Additionally, language has been included in the results section (Safety) explicitly stating that the trial did not stop early based on the definition of extreme toxicities and the Bayesian stopping rules. The extreme toxicity rates and 95% credible intervals have also been added.

2. The protocol proposed to use “A mixed model accounting for patient effects... to analyze

the longitudinal data on immunological values over time". The paper didn't include this analysis

Authors' response: Since we collected pre- and post-treatment samples from only 12 patients (7 in the tremelimumab arm and 5 in the cryoablation+tremelimumab arm) with only 1 post-treatment time point, the sample size was not suitable for mixed modeling analyses.

3. Though the sample size is small, OS and PFS curves showed clear separation between two arms, which is good but there is not statistical testing (log-rank test) or modeling (cox model with HR and 95% confidence interval) results.

Authors' response: We thank the Reviewer for pointing this out. Since this is a non-comparative study, we added hazard ratios and 95% confidence intervals, but no p-values.

4. The study conducted many two group comparisons, and didn't mention any multiplicity adjustment method, such as Bonferroni or FDR.

Authors' response: We thank the Reviewer for bringing this to our attention. We have added the following sentence in the methods section: "P values were adjusted for multiple comparisons using the Benjamini-Hochberg Procedure" and included the adjusted p-values throughout, in the results section. Our conclusions do not change based on the adjusted p-values.

Some minor comments:

1. What is the barplot for Figure 3. B? It didn't show.

Authors' response: We apologize for the blank figure. We have addressed the Mac to PC issue and the figure is visible now

2. Why total sample size is N=30 in Table 2 (and Supp Table 1)? Treme Alone N=14 and Treme+Cryo N=15, it cannot be added to 30...

Authors' response: In the results section the initial sentence has been modified to discuss the withdrawal of consent of one patient prior to receiving treatment.

Reviewers' Comments:

Reviewer #1:

Remarks to the Author:

I think the authors have adequately responded to comments pending the other statistical review. The abstract is better although the wording could use some refinement ie the sentences on immune cell infiltration towards the end are redundant.

Reviewer #2:

Remarks to the Author:

Thanks for the thorough response. My comments have been addressed.

Manuscript NCOMMS-21-17493: A pilot study of Tremelimumab with or without cryoablation in patients with metastatic renal cell carcinoma
Response to the Editor's Comments

REVIEWERS' COMMENTS

Reviewer #1 (Remarks to the Author):

I think the authors have adequately responded to comments pending the other statistical review. The abstract is better although the wording could use some refinement ie the sentences on immune cell infiltration towards the end are redundant.

Author's Response: We appreciate the reviewer's comments and we have modified the abstract and removed the redundant sentence from the end.

Reviewer #2 (Remarks to the Author):

Thanks for the thorough response. My comments have been addressed.

Author's Response: We thank the reviewer for the response.